# Collaborative Retrieval for Large Language Model-based Conversational Recommender Systems

Submission Id: 698

## Abstract

Conversational recommender systems (CRS) aim to provide personalized recommendations via interactive dialogues with users. While large language models (LLMs) enhance CRS with their superior understanding of context-based user preferences, they typically struggle to leverage behavioral data, which has proven to be the key for classical collaborative filtering approaches. For this reason, we propose CRAG—Collaborative Retrieval Augmented Generation for LLM-based CRS. To the best of our knowledge, CRAG is the first approach that combines state-of-the-art LLMs with collaborative filtering for conversational recommendations. Our experiments on two publicly available conversational datasets in the movie domain, i.e., a refined Reddit dataset as well as the Redial dataset, demonstrate the superior item coverage and recommendation performance of CRAG, compared to several CRS baselines. Moreover, we observe that the improvements are mainly due to better recommendation accuracy on recently released movies. The code is anonymously available at: https://anonymous.4open.science/r/CRAG-8CBE.

## CCS Concepts

• **Information systems → Recommender systems**.

## 1 Introduction

With the exponential growth of content on the Web, recommender system (RS) has become an indispensable component for digital service platforms [16]. Traditional RSs, such as collaborative filtering [18], have demonstrated effectiveness in leveraging historical user-item interactions for recommendations. Conversational recommender systems (CRS) offer a more engaging and interactive environment for users, which enables users to express their preferences freely and refine their vague thoughts through multiple rounds of natural language interactions [15, 30], resulting in more personalized and contextually aware recommendations.

Nevertheless, the fundamental challenge for CRS lies in the comprehensive modeling of entities (e.g., items) and context (e.g., non-item texts) in the user query, which is essential for both dialogue understanding and response generation (see Fig. 1 for an example). Early CRSs [20, 30] utilize traditional RS models, such as factorization machine [26] or denoising auto-encoder [32], and sequential models, such as recurrent neural network (RNN) [6], to separately model the entity and context. Subsequently, external entity/word-level knowledge graphs (e.g., DBpedia [2] and ConceptNet [27]) and pretrained transformers [31] were introduced to enrich the entity/context representations and support conversational recommendation generations [5, 8, 33, 41]. In addition, strategies such as cross-attention [5, 41], mutual information maximization [33], and contrastive learning [42] were introduced to fuse the entity and context semantic information, such that the generation of entity and context in the response are aligned in the same space.

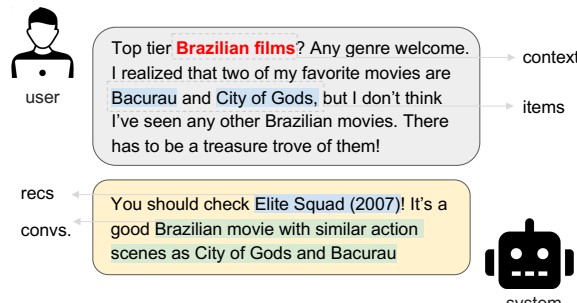

**Figure 1: An example for conversational recommendations, with entities and relevant context highlighted in user query.**

Recently, large language models (LLM), such as GPT-4o [23], Claude 3.5-Sonnet [1], etc., have demonstrated unprecedented understanding of both entities and context in the form of natural language [39]. Pretrained on large corpora from various domains, LLMs can be viewed as unstructured knowledge databases that encompass extensive knowledge of entities and their relations [25]. For instance, Xi et al. [38] showed that item-relevant knowledge prompted out of LLMs is conducive to improving recommendation accuracy. In addition, with LLMs' strong reasoning ability, user preference can be derived from both entity and context to generate recommendations [34]. Based on the advances in recent LLMs, He et al. [13] showed that LLMs (e.g., GPT-4) are good zero-shot recommender systems that substantially improve performance over previous methods with augmented external/pretrained knowledge and carefully designed recommendation and conversation modules.

While state-of-the-art LLMs have extensive knowledge and reasoning capabilities, they typically lack the ability to leverage collaborative filtering (CF), a foundational and effective technique in traditional recommender systems. Since user-item interaction data are usually proprietary and non-natural language-based, they are usually not included in the corpora and are difficult for LLMs to utilize. However, CF remains crucial for effective recommendations, even for pretrained transformer-based methods [35, 40, 43]. In addition, even if external CF information can be augmented for LLM-based CRS, existing work shows that adding more external knowledge does not necessarily lead to better performance [10], as it can introduce noise that biases the behavior of LLMs. Therefore, it is especially challenging to utilize the CF information to effectively complement context and LLMs' inherent content knowledge.

As an aside, in a different line of work, LLMs have been used for improvements in the classical sequential collaborative filtering setting [36]: Most works focus on *white-box* LLMs, where the model weights are accessible to the researcher [3, 14, 17, 40, 43]. White-box LLMs are generally smaller in scale compared to large proprietary LLMs, which are typically much more powerful, both in terms of

their knowledge and reasoning capabilities. Due to the inaccessibility of model weights, however, combining CF with *black-box* LLMs is comparatively less explored [25, 34, 35, 37]. For a more detailed discussion, the reader is referred to the Appendix A.

In this paper, to improve upon zero-shot LLMs, i.e., the current state-of-the-art for conversational recommender systems (CRS) [13], we propose CRAG, i.e., Collaborative Retrieval Augmented Generation for LLMs (see Section 3). To the best of our knowledge, CRAG is the first approach that combines state-of-the-art, black-box LLMs with collaborative filtering for the scenario of *conversational recommendations*. In our experiments in Sections 4.3 and 4.4, we show that CRAG leads to improved recommendation accuracy on two publicly available conversational datasets on movie recommendations. We also provide several ablations studies to shed more light on the inner workings of CRAG in Sections 4.5 and 4.6. Apart from that, we also plan to make a refined version of the Reddit dataset [13] on movie recommendations publicly available, where the extraction of the movies mentioned in the conversations is substantially improved (see Section 4.1.1). We also show (see **Finding 1** in Section 4.1.1) that this improvement in extraction accuracy can have a considerable impact on the derived insights.

## 2 Problem Formulation

In this section, we formally define the problem of CRS studied in this paper. Let $\mathcal{U}$ denote the set of users and $\mathcal{I}$ the set of items. A conversation between a user and the CRS is denoted as $C = \{(u_t, s_t, \mathcal{I}_t)\}_{t=1}^T$, where at the $t$-th turn, $u_t \in \{\texttt{User}, \texttt{System}\}$ generate an utterance $s_t = (w_1, w_2, \ldots, w_{N_t})$, which is composed of $N_t$ tokens from the vocabulary $\mathcal{V}$. $\mathcal{I}_t$ denotes the set of items mentioned in $s_t$. We assume that users can freely mention any item from $\mathcal{I}$ in their query, but the system can only recommend items from a fixed catalog (e.g., available movies on a specific platform like Netflix, HBO, or Hulu). We use $Q \subseteq \mathcal{I}$ to denote the catalog of items available for recommendations. Here, we note that $\mathcal{I}_t$ is usually **not** annotated by the user and may be empty if no items are mentioned at the $t$-th turn. The CRS backbone is a black box LLM $\Phi$, with available interaction data $\mathbf{R} = \{0, 1\}^{|\mathcal{U}_r| \times |\mathcal{I}|}$ as an external collaborative filtering database. Users in $\mathcal{U}_r$ does not have to be the same as $\mathcal{U}$. $\mathbf{r}_{u \in \mathcal{U}_r} \in \{0, 1\}^{|\mathcal{I}|}$ denotes co-occurrence pattern of items, and is generally not included in the LLM training corpora.

The focus of this paper is mainly on the recommender part of CRS, which aims to generate a ranked list of items $\hat{\mathcal{I}}_k$ from the catalog $Q$ based on the historical dialogue $C_{:k-1} = \{(u_t, s_t, \mathcal{I}_t)\}_{t=1}^{k-1}$ and the available interaction data $\mathbf{R}$, such that $\hat{\mathcal{I}}_k$ best matches the groundtruth items in $\mathcal{I}_k$ (if $\mathcal{I}_k \neq \emptyset$ and $u_k = \texttt{System}$).

## 3 Approach

In this section, we introduce CRAG, a collaborative retrieval-augmented LLM-based CRS with **two-step reflection**. CRAG consists of three main components, which are outlined in detail in the subsections below (see also Fig. 2): *(i) LLM-based entity linking*: This extracts items and user's attitude associated with each item mentioned in the dialogue and links them to the item database $\mathcal{I}$; *(ii) collaborative retrieval with context-aware reflection*: The extracted items are used as input to an adapted collaborative filtering (CF) model for item retrieval, which are then fed into a context-aware

reflection module that prompts an LLM to judge their context relevancy; and *(iii) recommendation generation with reflect-and-rerank*: an LLM is prompted to generate the recommendations based on the collaborative retrieval and then assign ordinal scores to all the items by their alignment with the dialogue, based on which they get reranked, resulting in the final list of recommended items.

While CRAG obviously hinges on the fact that there are some movies mentioned in the dialogue that can be extracted and then used for collaborative filtering, we can make this approach work in case there are no movies mentioned. Specifically, we replace step *(i)* with the one outlined in Section 3.4, where we simply ask an LLM to generate relevant movies based on the context of the dialogue.

### 3.1 LLM-based Entity Linking

*Entity linking*, i.e., extracting items $\mathcal{I}_k$ from the utterance $s_k$ and mapping them to the database $\mathcal{I}$, is crucial for CRS, as it bridges the gap between textual dialogues and external structured knowledge (e.g., knowledge graphs and interactions). However, existing methods, e.g., Bayesian models [7] or supervised finetuning of transformers [13], struggle with handling abbreviations, typos, and ambiguity, or rely on simulated data with seed items. Consequently, entity recognition noise is pervasive in the current CRS datasets[1].

#### 3.1.1 **LLM-based Entity Extraction**. In CRAG, we leverage the pretrained knowledge and reasoning ability of LLMs to extract the mentioned items in each utterance $s_t$. Additionally, we analyze the attitude associated with each item to capture the sentiment or stance context under which the item is mentioned by the user in the dialogue. This process (Fig. 2-*(i)*) can be formally denoted as:

$$\mathcal{I}_t^{raw} = f_e\left(\Phi\left(T_e, F_e, s_t\right)\right). \tag{1}$$

Here, $T_e$ is a task-specific prompt[2] to instruct the LLM $\Phi$ to reply with the standardized form of the items mentioned in utterance $s_t$ given that potential abbreviations, typos, and ambiguity could exist in $s_t$. In addition, to further improve the extraction efficiency, we design a *batch inference format instruction* $F_e$ to guide the LLM to reply with all the item-attitude pairs in utterance $s_t$ in the form of `"[item]<sep>[attitude]"`, where we empirically set `<sep>` to `"####"` as the dummy tokens that separate the item name and the associated attitude in the response. In $F_e$, we specifically instruct the LLM to output attitudes as numerical values in the range `{-2, -1, 0, 1, 2}`, representing attitude categories in the spectrum of {very negative, negative, neutral, positive, very positive}. This numerical encoding helps minimize errors in generation. With $F_e$, the raw set of item-attitude pairs $\mathcal{I}_t^{raw} = \left\{\left(i_{t,j}^{raw}, a_{t,j}\right)\right\}_j$ can be trivially extracted from the LLM's output using a string processing function $f_e$ that parses the lines and the `<sep>` tokens.

#### 3.1.2 **Bi-level Match and Reflection**. In the current stage, each raw item $i_{t,j}^{raw} \in \mathcal{I}_t^{raw}$ is a text string that may still contain (with a small probability) non-standardized forms, which the LLM might not fully correct during the extraction step. To accurately link the $i_{t,j}^{raw}$ to the item database $\mathcal{I}$, we introduce a bi-level match and reflection module that combines character-level and word-level fuzzy

---

[1] See Section B.1 for examples of noisy item recognitions for the Reddit dataset.
[2] The details of the prompts defined in the main paper are provided in Appendix C.

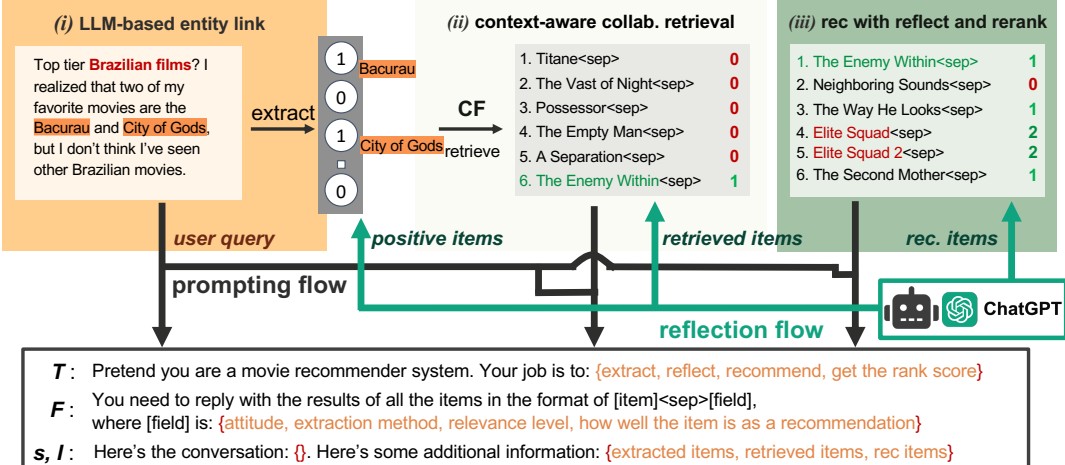

**Figure 2: Overview of CRAG for CRS and its three components: *(i)* LLM-based entity link, *(ii)* context-aware collaborative retrieval, and *(iii)* recommendation with reflect and rerank. The two reflection steps are emphasized in green arrows. The sub- and super-script for different task-specific prompt $T$, format instruction $F$, and item list $\mathcal{I}$ are omitted for simplicity.**

matching with an LLM-based reflection process to refine the entity linking accuracy. Specifically, character-level match addresses typos in $i_{t,j}^{raw}$ [13], whereas word-level match links certain abbreviations (e.g., "Star Wars I") to their full name in the database $\mathcal{I}$ (e.g., "Star Wars I - The Phantom Menace"). The above match processes produce two candidate sets denoted as $\mathcal{I}_t^{char}, \mathcal{I}_t^{word}$. Furthermore, we ask LLM to reflect on disagreements (if any) between $\mathcal{I}_t^{char}$ and $\mathcal{I}_t^{word}$, which is formally denoted as follows:

$$\mathcal{I}_t^{ref} = f_e^{ref}\left(\Phi\left(T_e^{ref}, F_e^{ref}, \mathcal{I}_t^{char}, \mathcal{I}_t^{word}, s_t\right)\right). \tag{2}$$

In this step, the task-specific prompt $T_e^{ref}$ instructs the LLM to reflect on the differences between $\mathcal{I}_t^{char}$ and $\mathcal{I}_t^{word}$ based on the utterance $s_t$. In addition, the *batch reflection format instruction* $F_e^{ref}$ guides the LLM to judge all the disagreements simultaneously and return the final results in the format "[match]<sep>[method]", where "[match]" is the item that the LLM determines to be correctly linked to the database $\mathcal{I}$ (could be empty if none is found), and "[method]" in {char, word, both, none} indicates the correct matching strategy. Finally, the function $f_e^{ref}$ processes the LLM's output by selecting, removing, or correcting each item based on the "[match]" and "[method]" fields to form the final set $\mathcal{I}_t^{ref}$.

## 3.2 Context-Aware Collaborative Retrieval

After extracting and linking items for each utterance $s_t$ in the dialogue $C_{:k-1}$ to the database $\mathcal{I}$, we introduce the collaborative retrieval module of CRAG. This module aims to retrieve context-relevant items based on the historical dialogue $C_{:k-1}$ and interaction data $\mathbf{R}$, which augments the query with collaborative filtering (CF) knowledge to enhance the LLM-based recommendations.

*3.2.1 **Collaborative Retrieval**.* Collaborative retrieval, similar to other retrieval-augmented generation (RAG) strategies [10], follows two main steps: query rewriting and similarity matching. The

overall process for collaborative retrieval is defined as follows:

$$\mathcal{I}_k^{CR} = \text{Top}_K\left(Sim\left(f_r\left(C_{:k-1}\right), Q; \mathbf{R}\right)\right), \tag{3}$$

where the query rewriting function $f_r(C_{:k-1})$ aggregates the positively mentioned items from the dialogue history $C_{:k-1}$, i.e., $\mathcal{I}_k^q = \cup_1^{k-1}\mathcal{I}_t$, and converts it into a multi-hot variable $\mathbf{r}_k \in \{0,1\}^{|\mathcal{I}|}$. Since it is generally risky to extrapolate negatively mentioned items through collaborative filtering (as the reason for disliking an item tends to be more subjective than collaborative in nature) and because of the small number of negative item mentions in the dialogues, we exclude the negatively mentioned items from the collaborative retrieval model. Afterward, we retrieve the top-$K$ items from the catalog $Q$ based on their similarity (via the $Sim$ function) with the items in $\mathcal{I}_k^q$ derived from the collaborative information.

Various CF methods [21, 22] can be used to learn the $Sim$ function based on the interaction matrix $\mathbf{R}$. In this paper, we utilize a simple while effective adapted EASE [29] objective as follows:

$$\min_{\mathbf{W}} \quad \|\mathbf{R}_Q - \mathbf{R}\mathbf{W}\|_F^2 + \lambda \cdot \|\mathbf{W}\|_F^2$$
$$\text{s.t. } \mathbf{W}_{i,j} = 0, \forall i = \text{ReID}(j), \tag{4}$$

where $\mathbf{R}_Q$ selects the columns in $\mathbf{R}$ that correspond to the items in the catalog $Q$, the asymmetric matrix $\mathbf{W} \in \mathbb{R}^{|\mathcal{I}| \times |Q|}$ maps the space of items that users mention freely in the dialogue (i.e., $\mathcal{I}$) to the space of items available for recommendation in the catalog $Q$, and the function ReID remaps the indices of catalog items from $\mathcal{I}$ to $Q$. The constraint in Eq. (4) prevents self-reconstruction from being used as a shortcut for the similarity matrix $\mathbf{W}$. Based on Eq. (4), the similarity function is then defined as $Sim(\mathcal{I}_q, Q) = \mathbf{r}_k^T \times \mathbf{W}$, which returns the similarity score of each item in $Q$ relative to the positively mentioned items in $C_{:k-1}$, i.e., $\mathcal{I}_k^q$. The scores are then used for collaborative retrieval. In addition, $\mathbf{W}$ is adjusted by more recent item-popularities based on the method introduced in [28].

*3.2.2* **Context-Aware Reflection**. Since collaborative retrieval defined in Eq. (4) does not consider the context information in the historical dialogue $C_{:k-1}$, directly augmenting the retrieved items $\mathcal{I}_k^{CR}$ in the user query as extra collaborative knowledge could introduce context-irrelevant information, thereby biasing the LLM's recommendations. To address this issue, we post-process the retrieved items via an LLM-based context-aware reflection step as:

$$\mathcal{I}_k^{aug} = f^{aug}\left(\Phi\left(T^{aug}, F^{aug}, C_{:k-1}, \mathcal{I}_k^{CR}\right)\right), \qquad (5)$$

where $T^{aug}$ is the task-specific prompt that instructs the LLM to reflect on the contextual relevancy of $\mathcal{I}_k^{CR}$ based on the historical dialogue $C_{:k-1}$. In addition, $F^{aug}$ is the *context-relevance batch reflection instruction* that guides the LLM to reply with the simultaneous judgment of all the items in $\mathcal{I}_k^{CR}$ in the format of `"[item]<sep>[relevance]"`, where `[relevance]` is a binary score in $\{0, 1\}$ indicating whether or not a retrieved `[item]` is contextually relevant. After the reflection, only context-relevant collaborative information is preserved in $\mathcal{I}_k^{aug}$, which is ready to be augmented into the query for generation. For example, in the example illustrated in Fig. 2-(ii), although all the retrieved movies are similar to `City of God` and `Bacurau`, all but `The Enemy Within` are Brazilian, where the rest are removed from $\mathcal{I}_k^{aug}$ after the reflection.

## 3.3 Recommendation with Reflect and Rerank

In the previous section, we focused on the *retrieval* phase of CRAG, where context-relevant collaborative information $\mathcal{I}_k^{aug}$ was obtained based on the historical dialogue and interaction data. In this section, we move to discuss the *generation* phase of CRAG, which generates the final recommendation list with LLM based on the collaborative retrieval $\mathcal{I}_k^{aug}$. This phase consists of three key steps: *(i)* collaborative query augmentation (pre-processing), *(ii)* LLM-based item generation, and *(iii)* reflect and rerank (post-processing).

*3.3.1* **Collaborative Query Augmentation**. The preliminary step of utilizing the retrieved collaborative knowledge $\mathcal{I}_k^{aug}$ is to augment it into the user query. This starts with adding a contextual note to emphasize the collaborative nature of the retrieved items, such as: `"Below are items other users tend to interact with given the positive items mentioned in the dialogue:"`. Afterward, $\mathcal{I}_k^{aug}$ is transformed into a textual string that lists the similarity-ranked items, with names separated by semi-colons.

We note that $\mathcal{I}_k^{aug}$ opens up to two interpretations in CRAG. From a RAG perspective, $\mathcal{I}_k^{aug}$ serves as extra CF information retrieved from an external user-item interaction database $\mathbf{R}$; from a recommendation perspective, $\mathcal{I}_k^{aug}$ also represents the possible item candidates that could be used in the final recommendations. Based on these interpretations, we design two distinct prompts to instruct the LLM on how to use the augmented collaborative information $\mathcal{I}_k^{aug}$: *(i)* a *rag* prompt that instructs the LLM to use the augmented information at its own discretion. *(ii)* a *rec* prompt that explicitly asks the LLM to consider the augmented items as candidates for recommendations (see Appendix C). Empirically, we find that different prompts work for different models. For example, GPT-4o enjoys the freedom in the *rag* prompt, whereas GPT-4 tends to ignore the retrieved $\mathcal{I}_k^{aug}$ under the same prompt and instead needs the *rec* prompt to force it to consider the items in $\mathcal{I}_k^{aug}$.

*3.3.2* **LLM-based Recommendations**. After constructing the collaborative augmented query $\mathcal{I}_{s,k}^{aug}$ from $\mathcal{I}_k^{aug}$ based on the previous part, it is appended to the historical dialogue $C_{:k-1}$ and input into the LLM to generate a preliminary recommendation list. The collaborative augmented generation step in CRAG is formalized as:

$$\mathcal{I}_k^{rec} = f^{rec}\left(\Phi\left(T^{rec}, F^{rec}, C_{:k-1}, \mathcal{I}_{s,k}^{aug}\right)\right), \qquad (6)$$

where the prompt $T^{rec}$ instructs the LLM to function as a conversational recommender system that responds based on both the historical dialogue $C_{:k-1}$ and the retrieved items $\mathcal{I}_{s,k}^{aug}$. The format instruction $F^{rec}$ guides the LLM to return $M$ standardized item names. Eq. (6) ensures that the generated recommendations take into account both the dialogue context and the collaborative information, thereby addressing two key limitations of zero-shot LLM-based recommendation systems: their reduced effectiveness for newer items and the lack of collaborative filtering capabilities.

*3.3.3* **Reflect and Rerank**. While the collaborative knowledge from $\mathcal{I}_{s,k}^{aug}$ substantially enhances the relevancy of generated recommendations, it can also trigger a bias inherent in LLMs, where the attention mechanism tends to replicate the order of items in the prompt. Since the item rank in $\mathcal{I}_{s,k}^{aug}$ only considers collaborative information, the most relevant items in $\mathcal{I}_k^{rec}$ generated by LLM (which are not necessarily in $\mathcal{I}_{s,k}^{aug}$) may not be ranked on the top.

Here, a naive approach to mitigate the bias is to directly ask the LLM to rerank the recommendations in $\mathcal{I}_k^{rec}$. However, this strategy usually does not work (i.e., results in missing items or nonsense reranked list), which is probably due to the large semantic gap between the input item list $\mathcal{I}_k^{rec}$ and the reranked output based on context relevancy. To bridge the semantic gap, we propose a reflect-and-rerank module in CRAG, which assigns ordinal scores to each item in $\mathcal{I}_k^{rec}$ based on how well it aligns as a recommendation with the dialogue history $C_{:k-1}$. This process is formalized as follows:

$$\mathcal{I}_k^{r\&r} = f^{r\&r}\left(\Phi\left(T^{r\&r}, F^{r\&r}, C_{:k-1}, \mathcal{I}_k^{rec}\right)\right), \qquad (7)$$

where the task-specific prompt $T^{r\&r}$ instructs the LLM to reflect on the recommendations and assign scores to all the items in $\mathcal{I}_k^{rec}$ based on $C_{:k-1}$. In addition, the *batch reflect-and-rerank instruction* $F^{r\&r}$ guides the LLM to return the scores for all the items in $\mathcal{I}_k^{rec}$ simultaneously in the format `"[item]<sep>[score]"`, where `[score]` $\in$ `"{-2, -1, 0, 1, 2}"` corresponds to the recommendation alignment in $\{$`very bad, bad, neutral, good, very good`$\}$. These scores serve as a reference for evaluating the relative suitability of each item, providing an intermediate step to address the semantic gap between the input items $\mathcal{I}_k^{rec}$ and the context-aware reranked items $\mathcal{I}_k^{r\&r}$. From the example in Fig. 2-(iii), we can see that even though `The Enemy Within` is a good recommendation based on the collaborative information, more relevant ones such as `"Elite Squad"` and `"Elite Squad 2"` are reranked on the top.

## 3.4 Conversations without Item Mentions

The previous sections address the scenario where the user has mentioned positive items in the historical dialogue $C_{:k-1}$. However, there is also the a small number of cases where the user has not mentioned any item in $C_{:k-1}$. To address this scenario without a

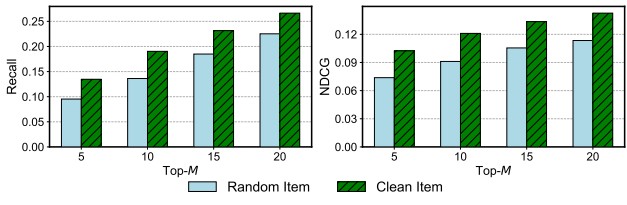

**Figure 3: Comparison of zero-shot LLM on Reddit-v2 dataset and the one with randomly replaced items.**

mentioned item, we first prompt the LLM to infer potential items the user might like based on $C_{:k-1}$. These generated items are then mapped to the item database $\mathcal{I}$ via the entity linking mechanism introduced in Section 3.1. These items can be treated as $\mathcal{I}_t$ in Eq. (4), and the remaining part of CRAG process remains the same.

## 4 Empirical Study

### 4.1 CRS Datasets

In this section, we introduce the established Reddit-v2 dataset and the public Redial dataset used for CRS model evaluations.

*4.1.1 **Reddit-v2 Dataset**.* The largest real-world CRS dataset is the Reddit dataset [13], which consists of dialogues collected from the Reddit website under movie-seeking topics. In each dialogue, the movie-seeker is treated as the user, whereas the responder is treated as the system. In addition, items (i.e., movies) were extracted from the utterances using a T5 model [24] fine-tuned on a simulated utterance-item dataset. However, due to the limited capacity of both T5 and the simulated training data, this suffered from rather low accuracy in the extracted movies (see Tables 2, 3 in the Appendix).

To address the issue, we first refine the Reddit dataset (which we name ***Reddit-v2***), where the items are extracted from the utterances with GPT-4o based on the batch inference with reflection strategy defined in Section 3.1. A qualitative comparison between the item extraction in Reddit-v2 and the original Reddit dataset is provided in Section B.1 in the Appendix. To quantitatively verify the effectiveness of the item extraction, we reproduce the item-replacement experiment in [13], which compares the performance of a Zero-shot LLM for CRS before and after randomly replacing the extracted items in the utterances. The results are illustrated in Fig. 3. In Fig. 3 we can see a noticeable degradation in the recommendation performance when items in the dialogue are randomly replaced (∼ 0.05 for recall@5). This challenges the claim in He et al. [13] that randomly replacing item names "has minor influence" on the zero-shot LLM-based CRS (where the difference of recall@5 is less than 0.01). These results not only demonstrate that the Reddit-v2 dataset is significantly cleaner, but also leads to **Finding 1**: *the items mentioned in the conversation play a critical role for LLMs to generate the recommendations*. To facilitate the comparison with knowledge graph (KG)-based CRS baselines, we map the extracted movie names to the DBPedia entities [2] and extract non-item entities that are within two hops of the item entities.

*4.1.2 **Redial Dataset**.* Another CRS dataset that we consider in this paper is the Redial dataset, which is crowd-sourced from Amazon Mechanical Turk (AMT) by Li et al. [20]. In the Redial dataset, items in the utterances are tagged by the Turkers alongside the

conversations. Although this requirement eliminates the necessity of entity recognition and its associated possible inaccuracies, this is not realistic in real-world applications. In addition, we find (as with He et al. [13]) that the conversations in the Redial data can be overly polite, rigid, and succinct (e.g., replying "Whatever Whatever I'm open to any suggestion." when being asked for preferences), which are comparatively poor in context information.

### 4.2 Experimental Setup

In Reddit-v2, we use the same subset drawn from the dialogues in the last month (i.e., *Dec. 2022*) as [13] as the test set, a subset from the month prior as the validation set, and all the dialogues from the prior months (before Oct. 2022) as the training set (see Fig. 11 in the Appendix for the distribution of dialogue start date). Additionally, we establish the interaction data **R** based on the Reddit-v2 training set, where each dialogue is treated as a pseudo-user $i$, and all positively mentioned items are treated as the historical interactions $\mathbf{r}_i$. For both Reddit-v2 and Redial datasets, we treat the set of mentioned items in the dialogues as the item database $\mathcal{I}$ and all items in the system responses as the catalog $Q$. The statistics of the test set for both datasets are shown in Table 1 in the Appendix.

We consider two LLMs, i.e., GPT-4 and the latest GPT-4o, as the backbone for CRAG[3]. We excluded other models, such as GPT-3.5 and GPT-3.5-turbo, due to their significantly weaker instruction-following capabilities. Here, we note that for CRS evaluation, the choice of LLM faces an inherent trade-off between item coverage and data-leakage risk. For GPT-4, approximately 15% of the movies in the item database $\mathcal{I}$ are released after its pretraining cut-off date, but all the test dialogues are after its cut-off date. In contrast, GPT-4o covers all the items, but the test dialogues are before its cut-off date. However, even for GPT-4o, the risk of data leakage is low, as Reddit closed its crawling interface to fight LLMs long before GPT-4o. In addition, since the strongest baseline, i.e., the zero-shot LLM proposed in [13], will use the same LLM as CRAG, the comparison still remains fair despite the trade-off in LLM selections.

### 4.3 Analysis of the Two-step Reflections

In our experiments, we first analyze the key contributions in CRAG, i.e., the two-step reflection process defined in Eqs. (5) and (7), which ensures contextual relevancy of retrieved items and reranks items in the final recommendation list to prioritize more relevant items. Specifically, we aim to explore when reflection works and how each reflection step contributes to the performance of CRAG.

*4.3.1 **Evaluation Setup**.* To answer the above research questions, we design two variants of CRAG, i.e., CRAG-nR12, CRAG-nR2, and explore their performance when the number of items in collaborative retrieval (i.e., parameter $K$ in Eq. (3)) increases. Specifically, in CRAG-nR12, we removed both reflection steps, whereas in CRAG-nR2 only the final reflect-and-rerank step is removed. We note that when $K = 0$, both CRAG-nR12 and CRAG-nR2 reduce to the zero-shot LLM proposed in [13]. Due to context-aware reflection, the number of items actually augmented into the query for recommendations could be less than $K$ for CRAG and CRAG-nR2. In the recommendation step, all three models are asked to recommend **20** movies.

---

[3]Due to space limitations, we only report CRAG with GPT-4o backbone in the main paper, while results with the GPT-4 backbone are provided in Appendix D.

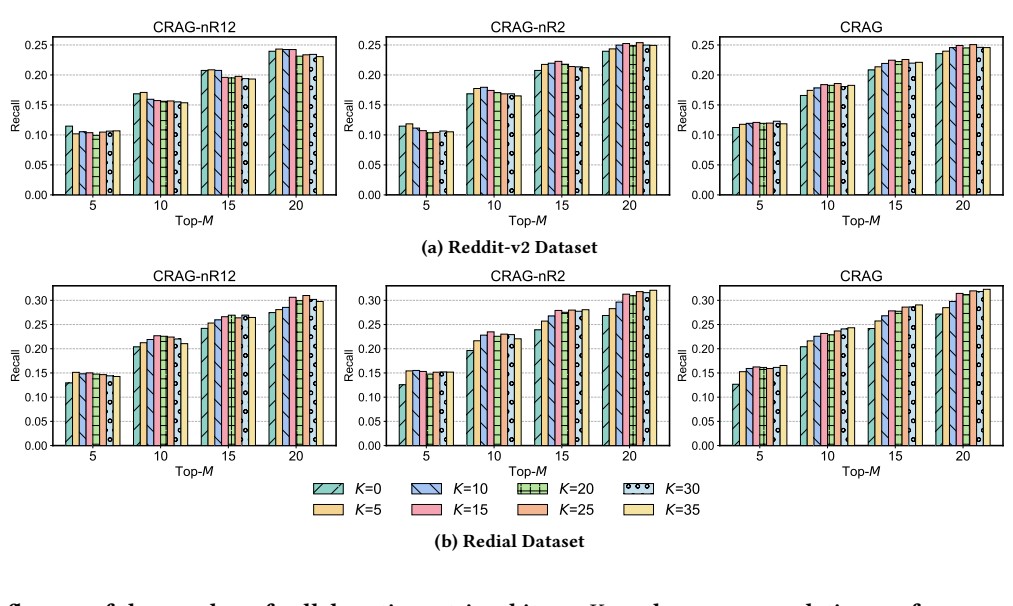

(a) Reddit-v2 Dataset

(b) Redial Dataset

**Figure 4: The influence of the number of collaborative retrieval items $K$ on the recommendation performance of CRAG-nR12, CRAG-nR2, and CRAG. X-axis denotes the recall evaluated at top-$M$ generated items, and different bars show different $K$.**

### 4.3.2 Intra-variant Comparisons.

We first consider each CRAG variant *separately* and focus on the trend of performance with varied $K$. The results are shown in Fig. 4, where the bar-group at top-$M$ shows the trend of recall@$M$ when $K$ increases from 0 to 35. In Fig. 4 we can see the following three interesting results:

**Finding 2.** *Naive collaborative retrieval is not very effective.* On the Reddit-v2 data, the performance of CRAG-nR12, i.e., the variant without any reflection, generally *decreases* when more items are retrieved and augmented into the query for all the $M$. Judged by recall@5, CRAG-nR12 even *degrades* the performance w.r.t. the zero-shot LLM for all the $K$. This makes sense because the raw retrieval, i.e., $\mathcal{I}_k^{CR}$, does not consider the context in historical dialogue, where context-irrelevant items bias the recommendations of the LLM.

**Finding 3.** *Context-aware reflection improves the coverage of relevant items but struggles with item rank.* This is reflected by the recall@20 bar group for CRAG-nR2, where the metric generally increases with a larger value of $K$. However, regarding recall@5 and 10 of CRAG-nR2, the metrics (quickly peak and then) decrease as more items $K$ are retrieved. This suggests that, with growing $K$, an increased number of relevant items are recommended in the top 20 positions, but not in the top 5 and 10 positions.

**Finding 4.** *Reflect-and-rerank addresses the rank bias and prioritizes most relevant items.* Regarding CRAG, also recall@5 and recall@10 increase with growing $K$ (besides recall@20, as also in CRAG-nR2). This suggests that the relevant items are ranked not only in the top-20, but also increasingly in the top-5 and 10 by CRAG.

The above findings lead to the conclusion that *LLMs are able to identify relevant items even if they cannot generate them.* This is exemplified by the fact that for $K > 0$, CRAG is able to exceed the performance of zero-shot LLMs (i.e., at $K = 0$), i.e., an increased number of relevant items is among the top-$M$. As the LLM reflects on additional items (which are predicted by collaborative filtering)

in CRAG, this result suggests that the LLM is able to identify relevant items from among these additional items, even though the LLM was not able to generate these additional relevant items itself.

### 4.3.3 Cross-variant Comparison.

In addition, we compare across the CRAG variants in Fig. 5, from which we can draw *two more interesting conclusions* that are not evident in Fig. 4:

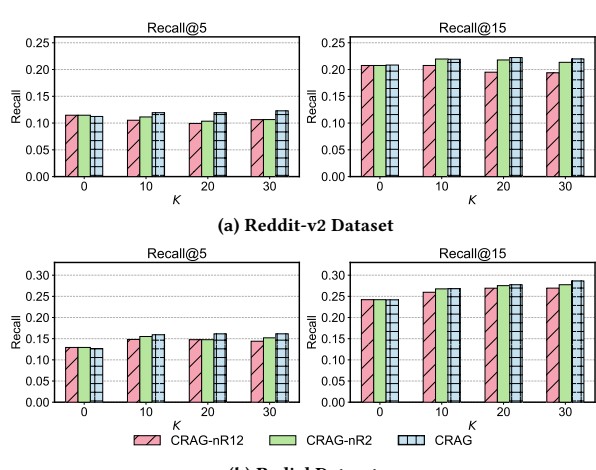

(a) Reddit-v2 Dataset

(b) Redial Dataset

**Figure 5: Comparison across different CRAG variants.**

**Finding 5.** *Self-reflection does not work.* We note that when $K = 0$, CRAG-nR12 and CRAG-nR2 degenerate to the zero-shot LLM, and CRAG degenerates to the model that adds self-reflection on zero-shot generations. The left-most bar group in Fig. 4 shows that when the recommendations are generated without external knowledge, self-reflection on the final recommendation list does not help. This makes sense, as for the zero-shot model, the items

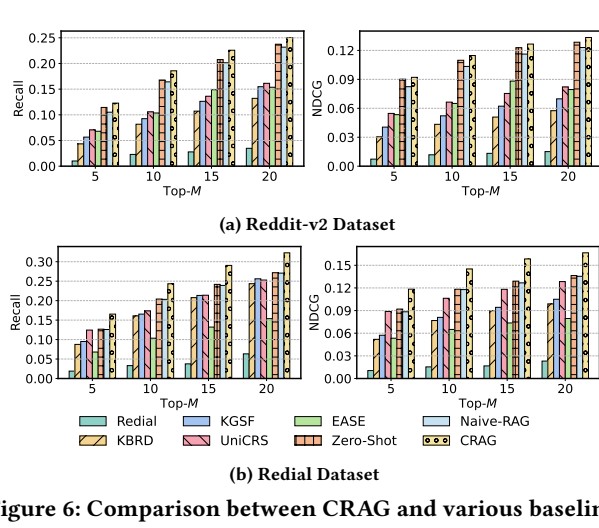

**(a) Reddit-v2 Dataset**

**(b) Redial Dataset**

**Figure 6: Comparison between CRAG and various baselines on the established Reddit-v2 and Redial datasets.**

reflected upon are generated based only on the LLM's internal knowledge, where the reflection cannot introduce new knowledge.

**Finding 6.** *Context is important for both reflection steps.* The larger improvement of CRAG over the variants on `Reddit-v2` dataset compared with `Redial` dataset shows that the two-step reflection has a larger impact on dialogues with richer context information (as in the `Reddit-v2` data). This shows that CRAG is an effective approach for combining collaborative filtering-based retrieval with the context-understanding of LLMs to improve LLM-based CRS.

### 4.4 Comparison to Baselines

In this section, we compare CRAG with various state-of-the-art RNN-, transformer-, and LLM-based CRS baselines as follows:

- **Redial** [20] leverages a denoising autoencoder to model the mentioned items and to generate recommendations, while an RNN is used to model and generate conversations.
- **KBRD** [5] introduces a relational GNN (RGNN) on the DBpedia knowledge graph (KG) to model entities, and optimize similarity between co-occurring tokens and entities to fuse semantics.
- **KGSF** [41] incorporates a word-level KG from ConceptNet to model the conversations, with mutual information maximization w.r.t. entity KG embeddings to fuse the entity information.
- **UniCRS** [33] introduces a pretrained transformer to capture the context information, with cross-attention [31] w.r.t. the entity KG embeddings (RGNN) used for semantic fusion.
- **Zero-shot LLM** [13] directly inputs the historical dialogue with task-specific prompt and format instruction for CRS without any retrieval from external knowledge database.
- **Naive-RAG** [19] denotes the model that retrieves item-related sentences from a database of movie plots and metadata based on semantic similarity between the query and sentences.

For `Redial`, KBRD, and KGSF, we follow the implementation from CRSLab, where we adapt the evaluation codes (which replicate each conversation multiple times such that each one has exactly one groundtruth) to make it consistent with CRAG. In addition, we limit the recommendations of all the baselines to items in the catalog $C$. Finally, we include **EASE** [29] as a non-CRS baseline, whose

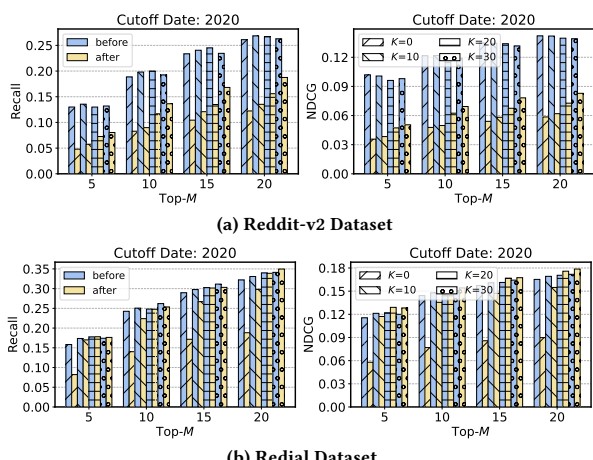

**(a) Reddit-v2 Dataset**

**(b) Redial Dataset**

**Figure 7: Comparison between the evaluations on test dialogues separated by the release year of the items (i.e., movies).**

recommendations are only based on the items mentioned in the dialogue (note that we adapt this model for collaborative retrieval).

The comparisons are shown in Fig. 6, where we can see that the `Redial` model, which separately models and generates items and conversations, achieves the lowest performance. KBRD and KGSF improve upon `redial` by using an external KG on entities, and introducing strategies to fuse the entity and context semantics in the dialogue. UniCRS further uses a pretrained transformer to model the context, which achieves the best performance among all the non-LLM-based baselines. However, due to the vast knowledge and reasoning ability of modern LLMs, Fig. 6 shows that a `Zero-shot` LLM improves substantially over the traditional methods. Regarding the RAG-based methods, we have the following findings:

**Finding 7.** Interestingly, we find that, `Naive-RAG`, which augments the `Zero-shot` LLM by retrieving relevant content/metadata as documents into the query, actually *degrades* in performance. The reason could be the large semantic gap between words in conversations and the implicit user preference. For example, for the dialogue illustrated in Fig. 2, most movie documents retrieved by `Naive-RAG` directly have Brazil/Brazilian in the movie name, but the user mentioned Brazilian only as a quantifier to his/her true preferences, i.e., movies similar to `City of God` and `Bacurau`.

**Finding 8.** CRAG achieves the best performance by all metrics across both datasets compared with both `Zero-Shot` LLM and `Naive-RAG`, which further demonstrates the effectiveness of the collaborative retrieval with two-step reflection in CRAG.

### 4.5 Evaluation w.r.t. Recency of Items

In this section, we shed some light on the main effect that we identified for CRAG: while CRAG improves the recommendation accuracy for all cases, the gains are more substantial for movies that were released more recently. This is corroborated by the following experiment: We first select a cut-off year (e.g., `2020`, but other years generally lead to similar results) and split the test dialogues into `before` and `after` groups: in the `before`-group, all groundtruth movies are released before the cut-off year, whereas in the `after`-group, at least one movie is released after the cut-off year. The results in Fig. 7 show the following interesting findings:

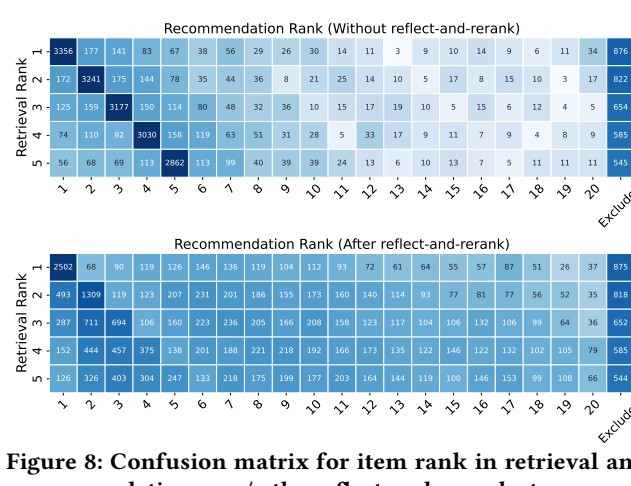

**Figure 8: Confusion matrix for item rank in retrieval and recommendation w, w/o the reflect-and-rerank step.**

**Finding 9.** *LLMs are less effective in recommending more recent items.* This is reflected by the overall lower performance of CRAG on the after-group (yellow bars) than the before-group (blue bars).

**Finding 10.** CRAG *leads to larger improvements for the recommendation of more recent items.* This is reflected by the larger metric increases for CRAG on the after-group as the number $K$ of items retrieved by collaborative filtering grows, compared to the before-group: visually, this is reflected by the steeper metric-improvements of the yellow bars (i.e., after-group) compared to the blue bars (i.e., before-group) when $K$ increases in Fig. 7.

## 4.6 Retrieval and Recommendation

Finally, we point out the importance of the reflect-and-rerank step of CRAG, see Section 3.3.3 and Fig. 2-*(iii)*. To this end, we examine the relation between the list of items retrieved after context-aware reflection, $\mathcal{I}_k^{aug}$, and the items $\mathcal{I}_k^{r\&r}$ that get actually recommended by CRAG: Fig. 8 shows the confusion matrix regarding the items in $\mathcal{I}_k^{aug}$ and the items in $\mathcal{I}_k^{r\&r}$ (to save space, we selected $K = 20$ as the example and only show the top 5 rows of the matrices). The matrix at the top of Fig. 8 shows the results before the reflect-and-rerank step, which leads to the following findings:

**Finding 11.** *LLMs have the bias to replicate the retrieved items without changing their order.* The dominating diagonal in the confusion matrix at the top of Fig. 8 (i.e., results before reflect-and-rerank) shows that the retrieved and recommended items tend to be in the same order. This implies that LLMs are indeed biased toward replicating retrieved items for recommendations in the same order.

**Finding 12.** *LLMs tend to replace irrelevant items in place instead of removing them and filling in the next ones.* Otherwise, given the large number of items excluded from recommendations, the confusion matrix at the top of Fig. 8 should have larger values for all lower-triangular elements (which denote the cases of upwardly lifted items from retrieval in the recommendations).

The confusion-matrix at the bottom of Fig. 8 shows that the reflect-and-rerank step in CRAG eliminates this bias of LLMs to replicate the retrieved items (as the dominating diagonal elements vanish) and prioritizes the relevant items towards the top of the list of $\mathcal{I}_k^{r\&r}$, irrespective if these items were retrieved from collaborative filtering or generated by the LLM in the earlier step of CRAG.

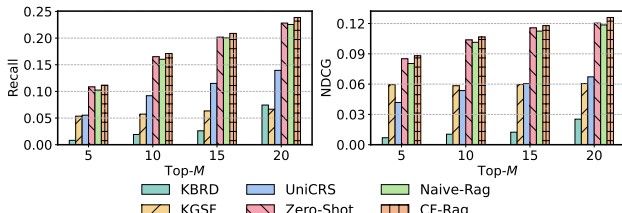

**Figure 9: Comparison of CRAG to the baselines on the conversations in the Reddit-v2 data without item mentions.**

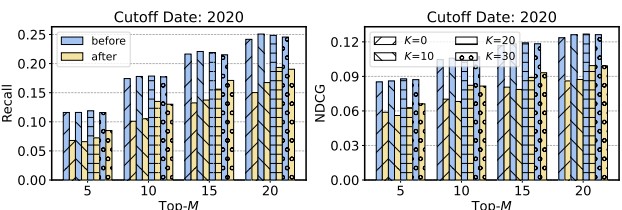

**Figure 10: Conversations on the Reddit-v2 data separated by the release year of the movie to be recommended.**

## 4.7 Conversations without Items Mentions

In this section, we provide results for CRAG on those conversations in which no items are explicitly mentioned, i.e., only context information is provided in the historical dialogue. We here focus on the Reddit-v2 data, as almost all the conversations in the Redial data contain at least one movie being mentioned. Comparison with baselines shown in Fig. 9 leads to **Finding 13**: *The relative performance of the baselines generally shows a similar trend as the case where items were mentioned in the conversation (see Fig. 6), although the improvement of* CRAG *over the* Zero-shot LLM *baseline is not as substantial.* Analogous to the experiments in Fig. 7, the results for the conversations where no movies are mentioned are shown in Fig. 10. This leads to **Finding 14**: *Despite the smaller overall improvement of* CRAG *over* Zero-shot LLM *in the case where no items are mentioned in the dialogue, the improvement in the recommendation of movies with more recent release-years is still very evident.* This again demonstrates the improvement of CRAG over LLMs in recommending movies that were released more recently.

## 5 Conclusions

In this paper, we proposed CRAG, Collaborative Retrieval Augmented Generation, the first approach that combines state-of-the-art, black-box LLMs with collaborative filtering for CRS to the best of our knowledge. In our experiments, we showed that this results in improved recommendation accuracy on two publicly available conversational datasets on movie recommendations, eclipsing the current state-of-the-art in conversational recommender systems, i.e., zero-shot LLMs. We also provided several ablations studies to shed more light on the inner workings of this approach. In particular, we found that recently released movies benefited especially from CRAG. Apart from that, we also establish a refined version of the publicly available Reddit dataset on movie recommendations, where the extraction of the movies mentioned in the conversations is greatly improved. We also showed that this improvement in extraction accuracy can have a considerable impact on the derived insights.

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

# Appendix

In the appendix, we discuss related work, provide detailed analysis and statistics of the established `Reddit-v2` dataset, and provide additional experimental results of `CRAG` with GPT-4 backbone.

## A  Related Work

In this section, we review the related work, which includes conversational recommender systems and LLM with collaborative filtering.

### A.1  Conversational Recommender Systems

Conversational recommender systems (CRS) aim to generate recommendations through natural language interactions with users [9, 15]. In CRS dialogues, there usually exist two types of information, i.e., *item* and *context*, where the latter denotes the non-item words that capture background information. To handle these two aspects, CRS models generally operate in two phases: *(i) modeling*, which learns to understand both entities and context in the dialogue, and *(ii) generation*, which generates items and natural language words based on the dialogue understanding as the response.

From the *modeling* perspective, a typical CRS involves three key components: entity modeling, context modeling, and semantic fusion. For entity modeling, various traditional recommender system models, such as factorization machines [26] and denoising autoencoders [32], have been employed to understand the mentioned items [5, 20]. Context modeling, on the other hand, often utilizes language models like recurrent neural networks (RNNs) [6] and transformers [12, 31] to capture the conversational flow and background information. To integrate entity and context information for a comprehensive understanding of the dialogue, semantic fusion techniques such as mutual information maximization [4] and cross-attention mechanisms [31] are employed [33, 41]. In addition, knowledge databases, such as DBPedia [2], ConceptNet [27], have been used to enhance dialogue modeling with external information.

For the *generation* phase, early methods introduced a switching mechanism, i.e., a binary predictor, to decide whether the next token should be a word or an entity [5, 20]. Afterward, approaches such as copy mechanism [11] are used to align items and tokens within the same generation space. Recently, Wang et al. [33] introduced an `<item>` token in context generation, enabling the system to recommend items based on the generated context seamlessly. *The advent of large language models (LLMs) has further blurred the boundaries between entity and context, as well as between modeling and generation phases of CRS.* LLMs possess extensive knowledge and reasoning abilities, allowing them to understand entities and context simultaneously. Moreover, the generation of items and context responses can be unified within the textual space, leveraging the LLM's capacity to produce coherent natural language outputs.

However, LLMs are comparatively less effective to recommend more recent items due to fewer relevant documents in the training corpora. In addition, LLMs struggle to leverage collaborative filtering knowledge, which are highly informative for recommendations.

### A.2  LLM with Collaborative Filtering

Recently, recommender system researchers have recognized the importance of integrating collaborative filtering (CF) with large language models (LLMs) to enhance recommendations [36]. Most works focus on the *white-box* LLMs, where the model weights are accessible to the researcher. One promising strategy is to introduce new tokens for users/items as soft prompts to capture the collaborative knowledge. These tokens can be independently assigned [3, 43] or clustered based on semantic indexing [14], and can be learned with language modeling on natural language sequences converted from user-item interactions [43], or predicted from pretrained CF models based on additional networks [17, 40]. White-box LLMs are generally smaller in scale compared to large proprietary LLMs. Due to the inaccessibility of model weights, combining CF with *black-box* LLMs is less explored. One strategy is to augment CF models with LLMs' analysis of user preferences [25, 34, 37]. To use the LLM itself as the recommender, Wu et al. [35] propose to transform user-item interactions into the prompt for LLMs to understand user preference and utilize a policy network to reduce redundancy. However, these approaches focus on traditional symmetric CF settings, which are not suitable for CRS with asymmetric item mention and recommendation and complex contextual information.

`CRAG` distinguishes itself by effectively combining CF with LLMs that leverage both interaction data and dialogue context. By introducing context-aware retrieval and a two-step reflection process, `CRAG` addresses the limitations of zero-shot LLM-based CRS and substantially enhances the recommendation quality.

## B  Details of the Reddit-v2 Dataset

In this section, we provide details of the established `Reddit-v2` dataset. Specifically, we provide qualitative analysis of movie name and attitude extraction, and various related dataset statistics.

### B.1  Comparison with Original Reddit Dataset

We first present the comparison results of movie name extraction between `Reddit-v2` and the original Reddit dataset in Tables 2 3. As shown above, `Reddit-v2` is more accurate in extracting relevant movie names, owing to its improved understanding of the context. Based on Tables 2, 3, we analyze the reasons why the original Reddit dataset fails to accurately extract movie names from user queries, which can be summarized into three cases as follows:

*(i)* First, we note that user queries are often noisy, with movies being misspelled or abbreviated. Without a complete understanding of the context, it becomes difficult for even a well-trained entity recognition model to accurately identify the correct movie names. For example, in the 501st example, the user query states:

> "...I feel like since the COVID lockdown I've seen every sci-fi action movie of this millennium... Things in the vein of the more modern AvP movies, Battle of LA, the Frank Grillo and his son fighting aliens series that I'm blanking on the name of, Pacific Rim franchise, etc."

In this instance, the user uses "AvP" to refer to `Alien vs. Predator`, yet the original Reddit system fails to extract the correct title.

*(ii)* In addition, we note that certain movie titles blend seamlessly into the context of the user query, making it challenging for the model to distinguish them from the natural language input. For example, in the 1092nd example, the user query states as follows:

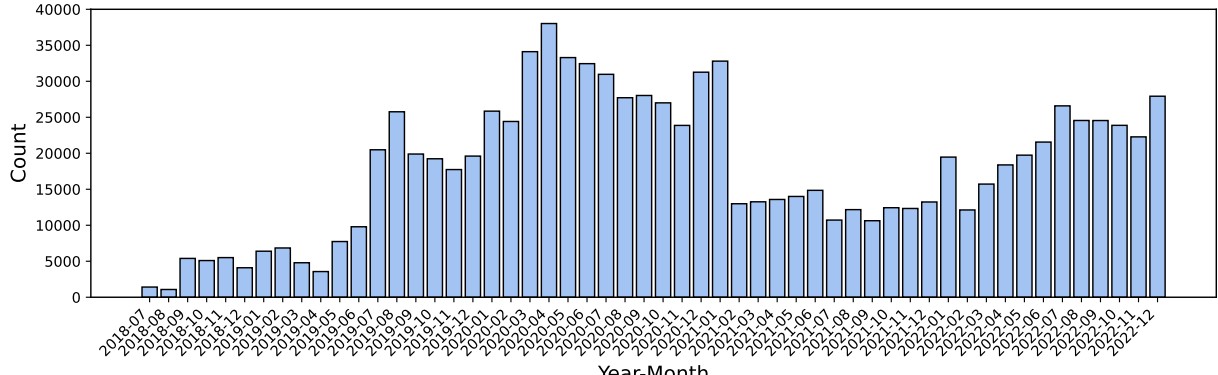

**Figure 11: Distribution of the start date of the dialogues in the *raw* Reddit-v2 dataset**

"Can you suggest some Netflix series for people who are really alone... For instance, I was watching the new Wednesday series and hoping I could relate to Wednesday Addams..."

Without prior knowledge of the series Wednesday, movie name extraction might mistakenly interpret "Wednesday" in the user query as a reference to a day of the week rather than the title of a show. This makes it challenging to correctly identify Wednesday in context. Similarly, in the 155th example, the query reads,

"...I have been looking for movies based on small American towns... The only movie that comes to mind is It (2017) ..."

Here, even without recognizing that It refers to a specific movie, the sentence remains semantically coherent. In both instances, accurate name extraction relies heavily on the language model's familiarity with the relevant movie or series titles.

*(iii)* Finally, we note that ambiguous answers may exist if the goal is simply to extract movie names from the database. In such cases, identifying the optimal solution relies heavily on the reasoning capabilities of the language model, which is typically achievable only by large language models. For example, in the 105th example in the Reddit-v2 test dataset, the user query states:

"...It gets mentioned a lot here, but Amelie is a movie that always lifts me up. This year I'd also recommend Everything, Everywhere, All at Once ."

The original Reddit dataset mistakenly recognizes three separate movies—Everything, Everywhere, and All at Once. However, based on the context, it is clear that the user is referring to the Oscar-winning film Everything Everywhere All at Once.

## B.2 Analysis of Attitude Extraction

We then qualitatively analyze the attitude extracted alongside the movie names in the Reddit-v2 dataset. The results are provided in Table 4, Table 5, and Table 6, which correspond to examples of positive, neutral, and negative attitudes from the user, respectively.

When users mention movies in their queries, they often convey a personal attitude toward them. In most cases, the LLM effectively infers whether the user holds a positive or negative sentiment toward the movies based on the surrounding context.

In the 519th example, the user query states:

"Best Foreign Movies? I recently watched Troll and Pan's Labyrinth . I wasn't always fond of movies with subtitles, but I really enjoy them now. What are some good Sci-fi/Fantasy foreign films?"

In this case, the LLM rates the user's attitude toward the two mentioned movies as a 2, indicating a positive sentiment. Another straightforward example is the 879th, where the user writes,

"...Movies like The Hangover, Superbad are just so stale and overrated. Any suggestions, please? I need a good laugh tonight."

Here, The Hangover and Superbad are rated as -2, reflecting the user's clearly negative attitude towards them.

If, in earlier stages of the conversation, movies are recommended but the user does not express any clear attitude toward them, they are assigned a rating of 0, indicating a neutral stance. For instance, in the 1418th example, the conversation goes as follows:

USER: [Request] Feel good movies?; SYSTEM: Rescued by Ruby ; USER: Gonna give this one a go right now, thanks!"

In this case, the LLM rates the user's attitude toward Rescued by Ruby as 0, reflecting the user's neutral attitude.

In cases where the user's attitude is mixed, the LLM can discern subtle nuances and read between the lines. For example, in the 1353rd example, the user writes the following in the query:

"Movies with interracial relationships, that aren't strictly ABOUT that? So not stuff like Jungle Fever, Get Out , etc."

Here, the users' attitudes toward Jungle Fever and Get Out are judged as -1, as the user does not express a strongly negative attitude but indicates that these movies do not align with their request.

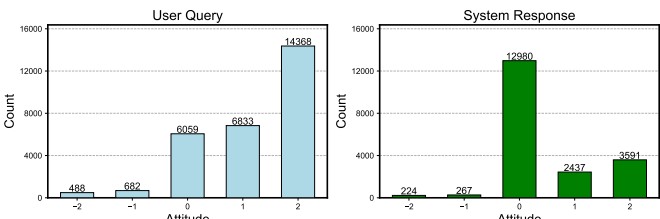

**Figure 12: Distribution of the LLM extracted attitudes in user and system response for Reddit-v2 test set.**

## B.3 Statistics of the Dataset

We run the LLM-based entity extraction introduced in Section 3.1 on all the conversations in the Reddit raw dataset established in [13], where the distribution of the start state of the dialogues are illustrated in Fig. 11. This dataset will be made publicly available afterward. For evaluation, we select the same subset (as [13]) of the dialogues with start date on the last month (i.e., *Dec. 2022*) as the test set for `Reddit-v2`, where the meta information (as well as the `Redial` dataset) is illustrated in Table 1. In addition, the distribution of attitudes for user query and system response are illustrated in Fig. 12. The large number of attitudes 0 for system response is due to succinct recommendations with only movie names, where the attitude is difficult to judge by LLM. Therefore, 0 is also treated as a positive attitude for system response in this paper.

**Table 1: Statistics of the testsets of Reddit-v2 and Redial in the main paper, where #Conv. (X) denotes the number of testing samples with no items mentioned in the dialogues.**

| Dataset | #Conv. | #Conv. (X) | #Items | #Catalog |
|---|---|---|---|---|
| Reddit-v2 | 5,613 | 2,231 | 5,384 | 4,752 |
| Redial [20] | 2,998 | 619 | 1,915 | 1,476 |

## C Prompts Defined in the Main Paper

In this section, we provide prompts we defined in the main paper.

### Eq. (1): LLM-based Entity Extraction

$T_e$: Pretend you are a movie recommender system. You (a recommender system) will be given a user's query that seeks movie recommendations. Based on the query, you need to extract movie names mentioned in the user's query and analyze the user's attitude toward each movie. You need to reply with standardized movie names (with grammatical errors corrected and abbreviations fixed), as well as the user's attitude toward the movie.

$F_e$: Specifically, the movie names need to be formatted in the IMDB style, with the year bracketed if possible (do not add the year if you are not sure). In addition, the attitude is represented in one of [-2, -1, 0, 1, 2], where -2 stands for very negative, -1 stands for negative, 0 stands for neutral, 1 stands for positive, and 2 stands for very positive. You need to reply with the number as an attitude instead of the textual

description. If there are movie names mentioned in the query, list each movie name and the user's attitude (number in -2 to 2) in the form of movie_name####attitude, where different movies are listed in different lines with no extra sentences. Reply NO if no movie names are mentioned in the query.

$s_t$: Here is the user's query: {}.

### Eq. (2): Reflection on Two-level Matched Entities

$T_e^{ref}$: Pretend you are a movie recommender system. You, as the recommender system, will be given part of the dialogue between a user seeking a movie recommendation and yourself, along with the extracted movie names (which may potentially be incorrect). Even if the extracted movie names are correct, the wording might not be precise. Therefore, you will be provided with the best match for each extracted movie name from an external database using (1) character-level fuzzy match and (2) word-level BM25 match (a space will be provided if no name can be found via the word-level match). Often, since these two matching methods focus on different levels of granularity, their results may not align. Based on the results, you must determine whether each movie name extraction is correct and what the precise movie name for that extracted name should be from the database.

$F_e^{ref}$: To reflect on this, for each extracted movie, you must respond with three terms separated by ####: (1) the raw movie name mentioned in the dialogue (raw refers to the exact text from the dialogue), (2) the precise movie name selected from fuzzy match or BM25 (reply with a space if the movie name extraction is incorrect or if neither match is precise), and (3) the correct extraction method, choosing from [fuzzy, BM25, none, both]. If the fuzzy match and BM25 results differ but both are probable, select the more probable one based on context as the correct name. List the reflection on each movie name in the exact form of raw _name####correct_name####method on a new line with no additional terms or sentences.

$s_t$: Here is the user's query: {}

$\mathcal{I}_t^{char}, \mathcal{I}_t^{word}$: Here are extracted movie names, fuzzy matches, and BM25 matches from the movie database in the form of extracted_name####fuzzy_match####BM25_match: {}

### Eq. (5): Reflection on Collaborative Retrieval

$T^{aug}$: Pretend you are a movie recommender system. I will give you a conversation between a user and you (a recommender system), as well as movies retrieved from the movie database based on the similarity with movies mentioned by the user in the context. You need to judge whether each retrieved movie is a good recommendation based on the context.

$F^{aug}$: You need to reply with the judgment of each movie in a line, in the form of movie_name####judgment, where judgment is a binary number 0, 1. Judgment 0 means the

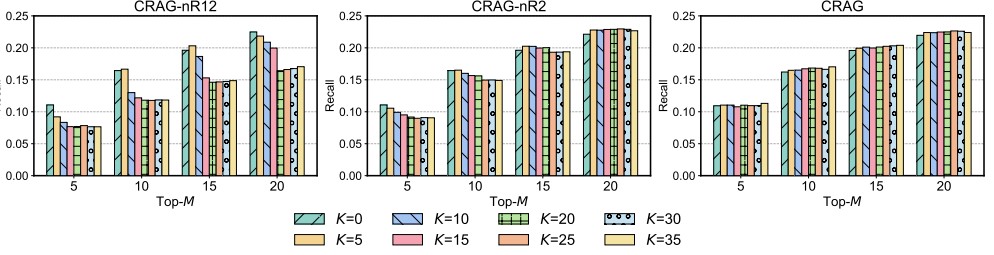

Figure 13: The influence of $K$ on CRAG-nR12, CRAG-nR2, and CRAG - results with GPT-4 backbone on the Reddit-v2 dataset.

movie is a bad recommendation, whereas judgment 1 means the movie is a good recommendation.

$C_{:k-1}$: Here is the conversation: {}.

$\mathcal{I}_k^{CR}$: Here are retrieved movies: {}

## Eq. (6): LLM-based Recommendations

$T^{rec}$: Pretend you are a movie recommender system. I will give you a conversation between a user and you (a recommender system). Based on the conversation, you need to reply with 20 movie recommendations without extra sentences.

$F^{rec}$: List the standardized title of each movie on a separate line.

$C_{:k-1}$: Here is the conversation: {}.

$\mathcal{I}_{s,k}^{aug}$: Based on movies mentioned in the conversation, here are some movies that are usually liked by other users: .

*rag* prompt (GPT-4o): Use the above information at your discretion (i.e., do not confine your recommendation to the above movies).

*rec* prompt (GPT-4): Consider using the above movies for recommendations."

## Eq. (7): Reflect and Rerank

$T^{r\&r}$: Pretend you are a movie recommender system. I will give you a conversation between a user and you (a recommender system), as well as some movie candidates from our movie database. You need to rate each retrieved movie as recommendations into five levels based on the conversation: 2 (great), 1 (good), 0 (normal), -1 (not good), -2 (bad).

$F^{r\&r}$: You need to reply with the rating of each movie in a line, in the form of movie_name####rating, where the rating should be an Integer, and 2 means great, 1 means good, 0 means normal, -1 means not good, and -2 means bad.

$C_{:k-1}$: Here is the conversation: {}

$\mathcal{I}_k^{rec}$: Here are the movie candidates: {}.

## D  Extra Experiments

In this subsection, we provide the experimental results of CRAG with GPT-4 as the backbone on the **Reddit-v2** dataset. Note that when

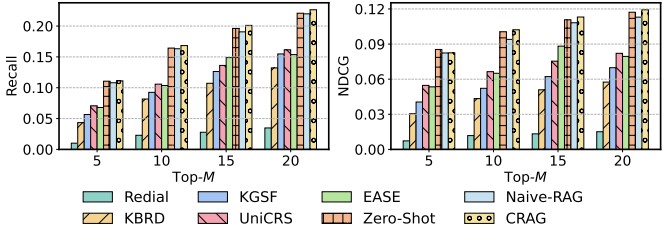

Figure 14: Comparison of CRAG (GPT-4) with baselines.

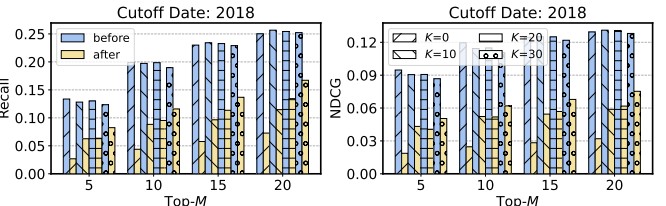

Figure 15: Evaluation of CRAG (GPT-4) w.r.t. item recency.

generating recommendations with Eq. (6), we use the *rec*-prompt instead of the *rag* prompt introduced in Section C.

### D.1  Analysis of the Two-step Reflections

We first run the experiments as with Section 4.3 of the main paper and summarize the results in Fig. 13. From the figure we find that the three CRAG variants follow the same trend, where *(i)* naive collaborative retrieval hurts metrics due to context-irrelevant information, *(ii)* introducing context-aware reflection improves the item coverage but struggles with item rank, *(iii)* and reflect-and-rerank leads to the prioritization of more relevant items on the top.

### D.2  Comparison with Baselines

We then compare the CRAG with GPT-4 backbone with the baselines The results are illustrated in Fig. 14, where the relative performance among the methods remains the same as with Fig. 6.

### D.3  Evaluation w.r.t. Item Recency

Finally, we evaluate the performance w.r.t. item recency. Since the cut-off date of GPT-4 is two years before GPT-4o, we set the cut-off year to 2018. From Fig. 15 we can find that, the improvement for CRAG with GPT-4 backbone over zero-shot model is largely due to the increased accuracy in recommendations of more recent items.

**Table 2: Comparison between Reddit-v2 and the original Reddit dataset for item extraction. The movie names that the original Reddit extracts incorrectly are marked in red. The evidence that supports our extraction in the user query is highlighted in both red and yellow boxes, where the red boxes denote the movies that the original Reddit dataset fails to extract.**

| Index | Context | Reddit-v2 | Original Reddit |
|---|---|---|---|
| 59 | ...i have watched 10 things i hate about you and its my absolute favorite, so im trying to find movies similar to 10 things i hate about you... | 1. 10 Things I Hate About You | **NONE** |
| 85 | ...Movies about exploration?. I love Master and Commander and I was thinking about movies about naval exploration...? Thanks' | 1. Master and Commander: The Far Side of the World | **NONE** |
| 155 | ...I have been looking for movies based on small american towns...The only movie that comes to my mind is It(2017) ... | 1. It | **NONE** |
| 156 | Revenge movies?. Looking for something like Kill Bill or John Wick. Would be very nice if it's on Netflix or Amazon Prime... | 1. Kill Bill: Vol. 2; 2. John Wick' | **1. Revenge; 2. Wild Bill;** 3. John Wick |
| 204 | Greatest cast in a movie?. I'd have to say Harlem nights ! Great movie, great cast and funny from start to finish! Eddie Murphy Richard Pryor Red foxx Arsenio Hall Charlie Murphy | 1. Harlem Nights | 1. Harlem Nights; **2. Red Fox** |
| 219 | Dream films. Inception is such a great film and I've not so much other films attempt a similar premise. So looking for those kinda films where people enter dreams or it has a dream-like state.' | 1. Inception; | **1. Dream Kiss;** 2. Inception |
| 243 | ...Some examples are: Last King of Scotland , A Bronx Tale, and Gangs of New York . I dunno why, but I love these types of films... | 1. The Last King of Scotland; 2. A Bronx Tale; 3. Gangs of New York | 1. A Bronx Tale; 2. Gangs of New York; **3. NONE** |
| 606 | Need movie like Eyes Wide Shut . Already watched Archive 81 that had masque secret society...Looking for movies about the wealthy elite like Rothchilds. | 1. Eyes Wide Shut; 2. Archive 81 | 1. Eyes Wide Shut; **2. Archive; 3. Archive; 4. Rothchild** |
| 639 | I am looking for every version of "A Christmas Carol" ever made.. Putting together a bit of a holiday film fest/challenge. I am looking for every version/adaptation of A Christmas Carol that has ever been made, from Scrooged to Muppets . | 1. Scrooged; 2. The Muppet Christmas Carol | **1. A Christmas Carol;** 2. Scrooged; **3. Puppets** |
| 710 | Out of nowhere Children's Horror?. I was just watching The Care Bears Movie (1985) and there is no way it can't be classified as Children's Horror. Is there any other unexpected horror in Children's IP?... | 1. The Care Bears Movie | 1. The Care Bears Movie; **2. Children's War** |

**Table 3: Comparison between Reddit-v2 and the original Reddit dataset for item extraction. The movie names that the original Reddit extracts incorrectly are marked in** red**. The evidence that supports our extraction in the system response is highlighted in both** red **and** yellow boxes **, where** red boxes **denote movies that the original Reddit dataset fails to extract.**

| Index | Response | Reddit-v2 | Original Reddit |
|---|---|---|---|
| 5 | Mermaids, Scent of a Woman, Mickey Blue Eyes, Mystic Pizza, and Rainy Day in NY | 1. Mermaids; 2. Scent of a Woman; 3. Mickey Blue Eyes; 4. Mystic Pizza; 5. A Rainy Day in New York | 1. Mermaids; 2. Scent of a Woman; 3. Mickey Blue Eyes; 4. Mystic Pizza; **5. NONE** |
| 9 | Easy, it's Warrior When About Today from The National starts playing at the end it just hits all of my feels | 1. Warrior | 1. Warrior; **2. About Adam** |
| 15 | Cocteau's 'Orpheus' it's like exactly what you're looking for You might also like Jarmusch's 'Paterson' and Van Sant's 'Drugstore Cowboy' and 'My Own Private Idaho' | 1. Orpheus; 2. Paterson; 3. Drugstore Cowboy; 4. My Own Private Idaho | 1. Orpheus; 2. Drugstore Cowboy; 3. My Own Private Idaho |
| 61 | Man bites dog , Martin and orloff, the doom generation | 1. Man Bites Dog; 2. Martin & Orloff; 3. The Doom Generation | 1. Martin & Orloff |
| 74 | Baise-moi    Shortbus Nymphomaniac Nymphomaniac 2 | 1. Baise-moi; 2. Shortbus; 3. Nymphomaniac: Vol. I; 4. Nymphomaniac: Vol. II | 1. Baise-moi; 2. Shortbus |
| 133 | You listed Conan , are you lumping Red Sonja into the Conan franchise. Just ensuring you haven't missed that one. | 1. Conan; 2. Red Sonja; | **1. Conman;2. Conman**; 3. Red Sonja |
| 159 | the harder they fall , it's on netflix also the crow | 1. The Harder They Fall; 2. The Crow; | 1. The Crow |
| 172 | The second and third Die Hard movies all take place within 24 hours as well. | 1. Die Hard 2; 2. Die Hard with a Vengeance | **1. Die Hard** |
| 105 | ...It gets mentioned a lot here but **Amelie** is a movie that always lifts me up. This year I'd also recommend ** Everything, Everywhere, All at Once **'... | 1. Amelie; 2. Everything Everywhere All at Once | 1. Amelie; **2. Everything; 3. Everywhere; 4. All at Once** |
| 207 | Gotta be It's a Mad, Mad, Mad, Mad World . | 1. It's a Mad Mad Mad Mad World; | **1. The Longest Day; 2. The Longest Day** |
| 269 | *North By Northwest* (1959). A bit like a Bond film before Bond. Hitchcock. Very stylish. Cary Grant and Eva Marie Saint. | 1. North by Northwest | 1. North by Northwest; **2. Bound; 3. Bound; 4. Bound; 5. Bound** |
| 308 | Lock, Stock, and Two Smoking Barrels. In Bruges. And There Were None (either the 1945 movie or the 2015 mini-series with Charles Dance). | 1. Lock, Stock and Two Smoking Barrels; 2. In Bruges; 3. And Then There Were None | **1. Lock; 2. Stuck; 3. Lock;** 4. In Bruges |

**Table 4: Examples of movies with positive attitude in Reddit-v2 dataset. The movie names are marked** green boxes **in the context or response.**

| Index | Context | Extracted movie names |
|---|---|---|
| 555 | ...Here is a list of movies that absolutely ruined me for weeks, some still haunt me with late night horror of being someone's victim simply because "You were home" 1. The Strangers; 2. Eden Lake; 3. Funny; Games; 4. Zodiac; 5. The Last House on the Left ... | 1. The Strangers; 2. Eden Lake; 3. Funny Games; 4. Zodiac; 5. The Last House on the Left |
| 500 | ...I feel like since the covid lockdown I've seen like every scifi action movie of this millenium...Things in the vein of the more modern AvP movies, Battle of LA , the Frank Grillo and his son fighting aliens series that I'm blanking on the name of, Pacific Rim franchise, etc... | 1. Alien vs. Predator; 2. Battle Los Angeles; 3. Pacific Rim |
| 519 | Best Foreign Movies?. I recently watched Troll and Pans Labyrinth . I wasn't always fond of movies with subtitles but I really enjoy them now. What are some good Sci-fi/Fantasy foreign films? | 1. Troll; 2. Pan's Labyrinth |
| 544 | Most Disturbing WW2 movies. Alright guys I saw all quiet on the western front the other night and I really enjoyed it. I'm looking for the most bloodiest war movie you can recommend me. Preferably WW2 | 1. All Quiet on the Western Front |
| 554 | Time loop movies. There are several great time loop movies out there, and some of my favorites include: Groundhog Day - In this classic comedy, a weatherman finds himself reliving ... to become a better person. Happy Death Day - A college student must relive the day of her murder over and over again until she figures out who the killer is. Edge of Tomorrow -... | 1. Groundhog Day; 2. Happy Death Day; 3. Edge of Tomorrow |
| 576 | I'm looking for movies with a global threat.. Specifically a movie where a bunch of organizations ... come together and work to understand, fight, and hopefully defeat it. The only example I can think of right now is " Contagion ". I greatly appreciate any and all suggestions :) Thank you! | 1. Contagion |
| 701 | ...I'm looking for something more where the movie's plot would go on and just display that the male's love interest or actress just happens to be older than him and that's it. An example of this is Water for Elephants where Reese Witherspoon is ten years older than Robert Pattinson, but the film still focuses on the circus storyline... | 1. Water for Elephants |
| 879 | the funniest non mainstream comedy.. I'm looking for a good comedy that I haven't seen before. I love comedy's like odd couple 2, palm springs, the wrong missy, vacation (2015),nothing to lose . Movies like the hang over, super bad are just so stale and overrated. Any suggestions please? I need a good laugh tonight. | 1. The Odd Couple II; 2. Palm Springs; 3. The Wrong Missy; 4. Vacation; 5. Nothing to Lose |
| 938 | ...Movies like Mean Girls and Freaky Friday ?. I really like these two movies. not particularly because of Lindsay btw although I liked her on these movies. are there like "go to movies" that are similar to these?... | 1. Mean Girls; 2. Freaky Friday |

**Table 5: Examples of movies with neutral attitude from the users in Reddit-v2. The movie names are marked with** yellow boxes **in the context or response.**

| Index | Context | Extracted movie names |
|---|---|---|
| 607 | What would you consider "must-see" movies?. I'm sorry if this has been asked a million and one times, I'm new here...Every time I look at lists of favorite movies, they always seem to be the same things, Citizen Kane, Shawshank, Godfather, Casablanca , etc. And no hate to those movies!! But they're classics for a reason, I've already seen them and want something new!... | 1. Citizen Kane; 2. The Shawshank Redemption; 3. The Godfather; 4. Casablanca |
| 683 | Movies about guns.. I'm seeking films about guns or involving lots of gun action. For example: Lord Of War Gun Crazy Hardcore Henry I am going to just fill the rest here for the mandatory text limit because I have nothing else to say. Please comment below. | 1. Lord of War; 2. Gun Crazy; 3. Hardcore Henry |
| 1035 | Akira (1988) Is an amazing film. Akira (1988), which I saw for the first time last night, completely floored me. I can't believe I haven't seen the film sooner after having it on my to-do list for so long. I'm not a huge anime fan Spirited Away and Pokémon are about the extent of my knowledge), but I think anyone would like this film... | 1. Spirited Away; 2. Pokémon |
| 1474 | ... I would like to see some movies where the main character or an important character is red haired, i don't mind if it's natural or not. Last movie i saw was Perfume: The Story of a Murderer and i was wondering why red haired/gingers women are so rare in movies. I would appreciate even movies where the girl is not the protagonist, tho keep in mind she should be on the screen more then 1 scene. Any type of movie is welcomed. Thank you in advance. | 1. Perfume: The Story of a Murderer |
| 1395 | My wife is currently getting a procedure done that will leave her face appearing severely burned for several days. Other than Nicolas Cage's Face/Off , what movies should I queue for our marathon while she recovers?... | 1. Face/Off |
| 1673 | Best of the Middle East. I had a chance to watch...I would love to see more great Egyptian/Middle Eastern/Arabic/North African films. Other than the Iranian ** A Girl Walks Home Alone At Night ** I haven't really seen much of anything from the region. Any suggestions on where to start?" | 1. A Girl Walks Home Alone at Night |
| 1690 | ...I'm asking this because I'm watching Thor: Love and Thunder for the first time and while it's not bad, it feels more like background noise or standard popcorn fare. It's fine and all but it got me thinking, what are some movies where my attention will be absolutely grabbed? Where pulling out my phone even to look at it for a second would be unwanted? | 1. Thor: Love and Thunder |
| 2903 | Sequels which pick up immediately from the original. What movies pick up exactly from where their originals leave off? I don't mean "a short while later" like Star Wars: A New Hope to The Empire Strikes Back , but straight shots with continuity... | 1. Star Wars: Episode IV - A New Hope; 2. Star Wars: Episode V - The Empire Strikes Back |

**Table 6: Examples of movies with negative attitude from the users in Reddit-v2. The movie names are marked** red boxes **in the context or response.**

| Index | Context | Extracted movie names |
|---|---|---|
| 590 | And please dont́ give me the shallow happy-go-lucky "Fundamentals of Caring" type of shit. I need deep, relatable emotions and metaphysical devastation. If I dont́ bawl at the screen questioning every Godś existance towards the end, it was not worth it. | 1. The Fundamentals of Caring |
| 701 | ...I am looking for a movie where a younger man and an older woman develop a romantic relationship...but it wouldn't be anything like The Graduate , or The Piano Teacher where their age gap is treated as taboo and is the centered plot... | 1. The Graduate; 2. The Piano Teacher |
| 879 | ...Movies like the hang over, super bad are just so stale and overrated. Any suggestions please? I need a good laugh tonight. | 1. The Hangover; 2.Superbad |
| 1061 | What's the best (bad) Christmas movie.. Bad Christmas movies are a guilty pleasure of mine...What are your favorite bad movies? Major studio release, or made for tv trash, I don't care. Just tell me the movie, who's in it, and a simple plot, if I haven't seen it, I'll go find it. No "good" movies though. Don't recommend White Christmas or " it's a wonderful life " not only do we all know them, but they are iconic... | 1. White Christmas; 2. It's a Wonderful Life |
| 1092 | Can you suggest some Netflix series that is for people who are really alone... For eg., I was watching the new Wednesday series and hoping that I could relate to Wednesday Addams, only to realize that it is just another teen drama where supposedly lonely and evil Wednesday Addams has multiple love interests and saves... | 1. Wednesday |
| 1131 | actually scary zombie/vampire movies?. I watched 28 Days Later which I've heard is scary but I found it rather boring. I also watched Braindead but it wasn't scary, just gross. As for vampire movies, I love them but I've never seen any that is actually scary to me. What do you think?... | 1. 28 Days Later; 2. BrainDead |
| 1160 | Intense romance with a happy and fulfilling ending.. I just watched King Kong (2006) and now I feel hollow inside. So sad. It's like an intense romance with a tragic ending so now I need an intense romance with an extremely fulfilling ending where the two lovers go through intense hardships... | 1. King Kong |
| 1335 | I'm looking for quality story sci-fi / fantasy from 2010-20s... What I mean is, i tried watch " Life " to find an fascinatic newer sci-fi, ended up being close to brutal and grotesque. I tried watching 4400 series, ended up being not that much about sci-fi but about trans/lesbian activism, teenage romance dramas, anti-christian activism... | 1. Life; 2. 4400 |
| 1353 | Movies with interracial relationships, that aren't strictly ABOUT that?. So not stuff like Jungle Fever , Get Out , etc. Films that could be in any genre, not just romance. The films can be I guess from any year, ideally in colour, but lean towards the '80s... | 1. Jungle Fever; 2. Get Out |

