# OpenReview forum: "Collaborative Retrieval for Large Language Model-based Conversational Recommender Systems"
_ACM.org/TheWebConf/2025/Conference — WWW 2025 Poster_

### Official Review · Reviewer_NoMF · 2024-11-16

**Novelty:** 4
**Technical Quality:** 4

**Review:**

The paper introduces CRAG, a new approach that combines large language models (LLMs) with collaborative filtering (CF) for conversational recommender systems (CRS). The experiments on the Reddit-v2 and Redial datasets demonstrate the potential of CRAG to improve recommendation performance. While the authors claimed that is the first work to apply the LLM with CF for CRS, the idea is not new as it was explored by several other works (but not CRS domain) such as [1],[2].

[1]Sun, Z., Si, Z., Zang, X., Zheng, K., Song, Y., Zhang, X. and Xu, J., 2024, October. Large Language Models Enhanced Collaborative Filtering. In Proceedings of the 33rd ACM International Conference on Information and Knowledge Management (pp. 2178-2188).

[2] Kim, S., Kang, H., Choi, S., Kim, D., Yang, M. and Park, C., 2024, August. Large language models meet collaborative filtering: An efficient all-round llm-based recommender system. In Proceedings of the 30th ACM SIGKDD Conference on Knowledge Discovery and Data Mining (pp. 1395-1406).

For the other weaknesses, please refer to the questions section.

**Questions:**

1. The authors mention that the current LLM in RS works are two lines which are black-box and white-box. The authors claim that the black-box LLM are less explored, but given the references provided, the black-box LLM seems more popular than white-box LLM as you are referencing five works for black-box but four works for white-box. I'm interested in how the authors can reach this conclusion. Meanwhile, this reason can not convince me why we must use the black-box LLM (i.e., ChatGPT in this work specifically)  In fact, many works conduct experiments on both (i.e., black box and white box).

2. It can not convince me that the author only conducted the experiments on ChatGPT and claimed that this paper combines the SOTA LLMs with CF. In addition to my previous point, I would like to see more experiments that use other more recent LLMs as backbone such as LLaMA-3.2, and Qwen-2.5. Only the use of the GPT family can not support your claim.

3. The selected baselines are not appropriate. Only the Zero-shot LLM can be treated as a proper baseline as the remaining are not LLM-based. More baselines in LLM-CF and LLM-based CRS are expected such as [1]-[4].

[1]Sun, Z., Si, Z., Zang, X., Zheng, K., Song, Y., Zhang, X. and Xu, J., 2024, October. Large Language Models Enhanced Collaborative Filtering. In Proceedings of the 33rd ACM International Conference on Information and Knowledge Management (pp. 2178-2188).

[2] Kim, S., Kang, H., Choi, S., Kim, D., Yang, M. and Park, C., 2024, August. Large language models meet collaborative filtering: An efficient all-round llm-based recommender system. In Proceedings of the 30th ACM SIGKDD Conference on Knowledge Discovery and Data Mining (pp. 1395-1406).

[3]Feng, Y., Liu, S., Xue, Z., Cai, Q., Hu, L., Jiang, P., Gai, K. and Sun, F., 2023. A large language model enhanced conversational recommender system. arXiv preprint arXiv:2308.06212.

[4]Sun, Z., Si, Z., Zang, X., Zheng, K., Song, Y., Zhang, X. and Xu, J., 2024, October. Large Language Models Enhanced Collaborative Filtering. In Proceedings of the 33rd ACM International Conference on Information and Knowledge Management (pp. 2178-2188)

4. Since the experiments are limited to the GPT family, I'm wondering about all of the LLM-related findings and claims' generalizability.

5. It is not clear to me regarding the motivation for the section 3.3.3. the authors say that collaborative knowledge will trigger a bias inherent in LLMs, and the rerank will not work properly. I'm wondering how the author observes that bias and the rerank does not work. Does any preliminary study or experiment support your claim?

6. Can you give me a specific example about section 3.4 where how the LLM is instructed to infer potential items? And how we can assess the performance of the proposed method in such conversations? How can the authors determine that the inferred potential items are correct and will not introduce extra noise to the model?

**Ethics Review Flag:**

Yes

**Reviewer Confidence:**

3: The reviewer is confident but not certain that the evaluation is correct

**Scope:**

3: The work is somewhat relevant to the Web and to the track, and is of narrow interest to a sub-community

---

### Official Review · Reviewer_9ezZ · 2024-11-26

**Novelty:** 4
**Technical Quality:** 5

**Review:**

Quality: The paper is well-researched and built upon the state-of-the-art by addressing a crucial limitation of conversational recommender systems (CRS) using large language models (LLMs): the inability to leverage collaborative filtering data. The paper is methodologically solid and well-supported by experiments, though the treatment of conversation that does not include items seems redundant.

Clarity: Clear structure and diagrams aid understanding.

Originality: The integration of collaborative filtering with LLMs in CRS is novel.

Significance: The work addresses key challenges in CRS, particularly the handling of recent data, and has strong practical and research implications, especially with open-source resources.

Pros:
- The narrative of the article is very rigorous, and the introduction of sufficient formulas makes the explanation clearer.
- The analysis provided in the experimental section is very comprehensive.
- The issue of data leakage is even considered.

Cons:
- In Figure 2, the arrows leading from (i) to (ii) and (iii) together show that the prompt was input three times. This is clear from the arrows’ bends and cross lines, but if the differences between the three inputs of (i) were indicated in the diagram, more information could be extracted directly from the figure.
- Section 3.4 seems like an unnecessary addition. If it is already possible to infer specific items from the user description, why is collaborative filtering still needed? Indeed, the method proposed in this paper is not applicable to conversations that do not mention any items (a cold-start scenario). The author acknowledges this and provides additional clarification, which is commendable.
- As a method combining LLMs and collaborative filtering for conversational recommendations, the baseline comparison does not include any conversational recommendation system based on LLMs. This weakens the persuasive power of the experimental results, as it does not sufficiently demonstrate the advantages of introducing collaborative filtering into conversational recommendation.

**Questions:**

I would like the author to explain why, if it is already possible to infer items from a conversation that does not mention them, this inferred item cannot be directly recommended to the user as the target, and instead, is treated as part of the interaction history to apply collaborative filtering for recommending other items.

**Reviewer Confidence:**

3: The reviewer is confident but not certain that the evaluation is correct

**Scope:**

4: The work is relevant to the Web and to the track, and is of broad interest to the community

---

### Official Review · Reviewer_N4SD · 2024-12-02

**Novelty:** 4
**Technical Quality:** 5

**Review:**

This paper introduces CRAG (Collaborative Retrieval Augmented Generation), a novel framework that enhances conversational recommender systems (CRS) by integrating black-box Large Language Models (LLMs) with collaborative filtering (CF). CRAG consists of three key components: (i) LLM-based entity linking, (ii) collaborative retrieval with context-aware reflection, and (iii) recommendation generation with reflect-and-rerank. This LLM-based CRS, employing a two-step reflection process, demonstrates significant improvements in recommendation accuracy, particularly for recently released movies, across publicly available datasets (Reddit-v2 and Redial). The authors provide well-documented code, ensuring strong support for reproducibility of the experiments. Additionally, they perform comprehensive ablation studies to assess the impact of various reflection mechanisms and different values of K. The paper is well-structured, with a clear problem formulation, methodology, and experimental analysis. The explanation of the two-step reflection process, along with its impact on recommendation ranking, is thorough and easy to follow.

**Questions:**

1. Have the authors explored or tested CRAG on datasets outside the movie domain? How might CRAG perform in other recommendation system (CRS) domains?
2. I notice that CRAG requires multiple (at least five) calls to GPT for a single recommendation, which may result in high costs and impact inference efficiency. Have the authors considered a trade-off between accuracy and efficiency, or are there specific modules that must rely on GPT for completion?
3. The paper mentions that CRAG includes two reflection steps: context-aware reflection on collaborative filtering items and reflect-and-rerank. However, I observe that there is also an entity-linking reflection step in the LLM-based entity linking process. Why not refer to it as a three-step reflection process? Additionally, I did not find an analysis of the entity-linking module in the experimental section.
4. I would like to understand how the model performs in low-resource scenarios. A significant portion of CRAG's performance gains comes from integrating collaborative filtering signals from the training set. In real-world settings, where high-quality collaborative signals may be scarce, can the model still perform effectively?
5. Figure 3 could potentially be clearer with some optimization. It might be helpful to refine the input and output sections and display the key intermediate components more clearly.

**Reviewer Confidence:**

3: The reviewer is confident but not certain that the evaluation is correct

**Scope:**

4: The work is relevant to the Web and to the track, and is of broad interest to the community

---

### Official Review · Reviewer_TUSb · 2024-12-02

**Novelty:** 5
**Technical Quality:** 5

**Review:**

The paper introduces a novel framework for integrating collaborative filtering techniques with state-of-the-art LLMs in conversational recommender systems. The authors propose a two-step reflection process to improve recommendation accuracy and contextual relevance, addressing limitations in traditional collaborative filtering and LLM-only systems.

### Pros
- CRAG effectively bridges collaborative filtering and LLMs, leveraging the strengths of both paradigms.
- The paper provides extensive ablation studies, analyzing the impact of key components such as the two-step reflection process.
- The introduction of the Reddit-v2 dataset, with cleaner item extraction, enhances the reproducibility and usability of CRS research.

### Cons
- The computational overhead of the two-step reflection process and collaborative filtering integration may limit its applicability to large-scale systems.
- The paper lacks some implementation details, such as the specific versions of GPT-4 and GPT-4o used, as well as settings like temperature.
- For reproducibility, it is recommended that the authors conduct experiments on open-source LLMs such as LLAMA or Mistral.
- The paper overuses bar charts, making it difficult to discern specific performances. It is suggested that key experiments be presented in tables for clarity.

**Questions:**

Could the authors provide more details about the computational costs of the two-step reflection process? Could this process introduce latency in real-time recommendation scenarios, and how might this be mitigated?

**Reviewer Confidence:**

3: The reviewer is confident but not certain that the evaluation is correct

**Scope:**

4: The work is relevant to the Web and to the track, and is of broad interest to the community

---

### Official Review · Reviewer_KAJq · 2024-12-02

**Novelty:** 6
**Technical Quality:** 6

**Review:**

This paper presents a novel approach that integrates large language models with collaborative filtering techniques to enhance conversational recommender systems. The authors demonstrate the effectiveness of CRAG through experiments on two publicly available datasets in the movie domain.

Pros:
1. CRAG is the first method to combine LLMs with collaborative filtering for conversational recommendations.
2. The authors provide extensive empirical results, including ablation studies that clarify the contributions of various components of the model, enhancing the understanding of its effectiveness.

Cons:
1. The paper lacks qualitative user studies or feedback mechanisms to assess user satisfaction with the recommendations, which is crucial for evaluating the real-world effectiveness of the system.

**Questions:**

The same as above.

**Reviewer Confidence:**

3: The reviewer is confident but not certain that the evaluation is correct

**Scope:**

4: The work is relevant to the Web and to the track, and is of broad interest to the community